# The Autoencoding Variational Autoencoder

**A. Taylan Cemgil**
DeepMind

**Sumedh Ghaisas**
DeepMind

**Krishnamurthy Dvijotham**
DeepMind

**Sven Gowal**
DeepMind

**Pushmeet Kohli**
DeepMind

## Abstract

Does a Variational AutoEncoder (VAE) consistently encode typical samples generated from its decoder? This paper shows that the perhaps surprising answer to this question is 'No'; a (nominally trained) VAE does not necessarily amortize inference for typical samples that it is capable of generating. We study the implications of this behaviour on the learned representations and also the consequences of fixing it by introducing a notion of self consistency. Our approach hinges on an alternative construction of the variational approximation distribution to the true posterior of an extended VAE model with a Markov chain alternating between the encoder and the decoder. The method can be used to train a VAE model from scratch or given an already trained VAE, it can be run as a post processing step in an entirely self supervised way without access to the original training data. Our experimental analysis reveals that encoders trained with our self-consistency approach lead to representations that are robust (insensitive) to perturbations in the input introduced by adversarial attacks. We provide experimental results on the ColorMnist and CelebA benchmark datasets that quantify the properties of the learned representations and compare the approach with a baseline that is specifically trained for the desired property.

## 1 Introduction

The variational AutoEncoder (VAE) is a deep generative model [10, 15] where one can simultaneously learn a decoder and an encoder from data. An attractive feature of the VAE is that while it estimates an implicit density model for a given dataset via the decoder, it also provides an amortized inference procedure for computing a latent representation via the encoder. While learning a generative model for data, the decoder is the key object of interest. However, when the goal is extracting useful features from data and learning a good representation, the encoder plays a more central role [20]. In this paper, we will focus primarily on the encoder and its representation capabilities.

Learning good representations is one of the fundamental problems in machine learning to facilitate data-efficient learning and to boost the ability to transfer to new tasks. The surprising effectiveness of representation learning in various domains such as natural language processing [7] or computer vision [11] has motivated several research directions, in particular learning representations with desirable properties like adversarial robustness, disentanglement or compactness [1, 3, 4, 5, 12].

In this paper, our starting point is based on the assumption that if the learned decoder can provide a good approximation to the true data distribution, the exact posterior distribution (implied by the decoder) tends to possess many of the mentioned desired properties of a good representation, such as robustness. On a high level, we want to approximate properties of the exact posterior, in a way that supports representation learning.

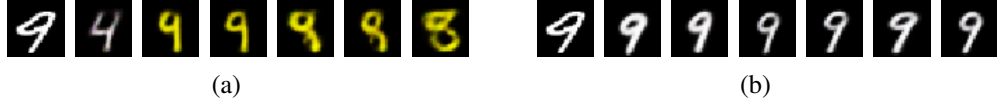

<div align="center">(a)          (b)</div>

Figure 1: Iteratively encoding and decoding a color MNIST image using a decoder and encoder (a) fitted a VAE with no observation noise (b) with AVAE. We find the drift in the generated images is an indicator of an inconsistency between the encoder and the decoder.

One may naturally ask in what extent this goal is different from learning a standard VAE, where the encoder is doing amortized posterior inference and is directly approximating the true posterior. As we are using finite data for fitting within a restricted family of approximation distributions using local optimization, there will be a gap between the exact posterior and the variational approximation. We will illustrate that for finite data, even global minimization of the VAE objective is not sufficient to enforce natural properties which we would like a representation to have.

We identify the source of the problem as an inconsistency between the decoder and encoder, attributed to the lack of *autoencoding*, see also [2]. We argue that, from a probabilistic perspective, autoencoding for a VAE should ideally mean that samples generated by the decoder can be consistently encoded. More precisely, given any typical sample from the decoder model, the approximate conditional posterior over the latents should be concentrated on the values that could be used to generate the sample. In this paper, we show (through analysis and experiments) that this is not the case for a VAE learned with normal training and we propose an additional specification to enforce the model to autoencode, bringing us to the choice of the title of this paper.

**Our Contributions:** The key contributions of our work can be summarized as follows:

- We uncover that widely used VAE models are not autoencoding - samples generated by the decoder of a VAE are not mapped to the corresponding representations by the encoder, and an additional constraint on the approximating distribution is required.

- We derive a novel variational approach, Autoencoding VAE (AVAE), that is based on a new lower bound of the true marginal likelihood, and also enables data augmentation and self supervised density estimation in a theoretically principled way.

- We demonstrate that the learned representations achieve adversarial robustness. We show robustness of the learned representations in downstream classification tasks on two benchmark datasets: colorMNIST and CelebA. Our results suggest that a high performance can be achieved without adversarial training.

## 2 The Variational Autoencoder

The VAE is a latent variable model that has the form

$$Z \sim p(Z) = \mathcal{N}(Z; 0, I) \qquad\qquad X|Z \sim p(X|Z, \theta) = \mathcal{N}(X; g(Z; \theta), vI) \quad (1)$$

where $\mathcal{N}(\cdot; \mu, \Sigma)$ denotes a Gaussian density with mean and covariance parameters $\mu$ and $\Sigma$, $v$ is a positive scalar variance parameter and $I$ is an identity matrix of suitable size. The mean function $g(Z; \theta)$ is parametrized typically by a deep neural network with parameters $\theta$ and the conditional distribution is known as the decoder. We use here a conditionally Gaussian observation model but other choices are possible, such as Bernoulli or Poisson, with mean parameters given by $g$.

To learn this model by maximum likelihood, one can use a variational approach to maximize the evidence lower bound (ELBO) defined as

$$\log p(X|\theta) \geq \langle \log p(X|Z, \theta) \rangle_{q(Z|X, \eta)} + \langle \log p(Z) \rangle_{q(Z|X, \eta)} + H[q(Z|X, \eta)] \equiv \mathcal{B}(\theta, \eta) \ (2)$$

where $\langle h \rangle_q \equiv \int h(a)q(a)da$ denotes the expectation of the test function $h$ with respect to the distribution $q$, and, $H$ is the entropy functional $H[q] = -\langle \log q \rangle_q$. Here, $q$ is an instrumental distribution, also known as the encoder, defined as

$$q(Z|X, \eta) = \mathcal{N}(Z; f^\mu(X; \eta), f^\Sigma(X; \eta))$$

Here, the functions $f^\mu$ and $f^\Sigma$ are also chosen as deep neural networks with parameters $\eta$. Using the reparametrization trick [10], it is possible to optimize $\theta$ and $\eta$ jointly by maximization of the ELBO using stochastic gradient descent aided by automatic differentiation. This mechanism is known as amortised inference, as once an encoder-decoder pair is trained, in principle the representation for a new data point can be readily calculated without the need of running a costly iterative inference procedure.

The ELBO in (2) is defined for a single data point $X$. It can be shown that in a batch setting, maximization of the ELBO is equivalent to minimization of the Kullback-Leibler divergence

$$\mathcal{KL}(\mathcal{Q}|\mathcal{P}) = \langle \log(\mathcal{Q}/\mathcal{P}) \rangle_\mathcal{Q} \qquad \mathcal{Q} = \pi(X)q(Z|X,\eta) \qquad \mathcal{P} = p(X|Z,\theta)p(Z) \quad (3)$$

with respect to $\theta$ and $\eta$. Here, $\pi$ is ideally the true data distribution, but in practice it is replaced by the dataset, i.e., the empirical data distribution $\hat\pi(X) = \frac{1}{N}\sum_i \delta(X - x^{(i)})$. This form has also an intuitive interpretation as the minimization of the divergence between two alternative factorizations of a joint distribution with the desired marginals. See [21] for alternative interpretations of the ELBO including the one in (3). In the sequel, we will build our approach on this interpretation.

## 2.1 Extended VAE model for a pair of observations

In this section, we will justify the VAE from an alternative perspective of learning a variational approximation in an extended model, essentially first arriving at an identical algorithm to the original VAE. This alternative perspective enables us to justify additional specifications that a consistent encoder/decoder pair should satisfy.

Imagine the following extended VAE model for a pair of observations $X, X'$

$$(Z, Z') \quad \sim \quad p_\rho(Z, Z') = \mathcal{N}\left( \left( \begin{array}{c} Z \\ Z' \end{array} \right) ; \left( \begin{array}{c} 0 \\ 0 \end{array} \right), \left( \begin{array}{cc} I & \rho I \\ \rho I & I \end{array} \right) \right) \quad (4)$$

$$\begin{aligned} X|Z &\quad \sim \quad p(X|Z,\theta) = \mathcal{N}(X; g(Z;\theta), vI) \\ X'|Z' &\quad \sim \quad p(X'|Z',\theta) = \mathcal{N}(X'; g(Z';\theta), vI) \end{aligned} \quad (5)$$

where the prior distribution $p_\rho(Z', Z)$ is chosen as symmetric with $p(Z') = p(Z)$, that is both marginals (here unit Gaussians) are equal in distribution. Here, $I$ is the identity matrix and $\rho$ is a hyperparameter $|\rho| \le 1$ that we will refer as the coupling strength. Note that we have $p_\rho(Z'|Z) = \mathcal{N}\left(Z'; \rho Z, (1 - \rho^2)I\right)$. The two border cases $\rho = 1$ and $\rho = 0$, correspond to $Z' = Z$, and, independence $p(Z', Z) = p(Z')p(Z)$ respectively. This model defines the joint distribution

$$\bar{\mathcal{P}} \equiv p(X'|Z';\theta)p(X|Z;\theta)p_\rho(Z', Z) \quad (6)$$

**Proposition 2.1.** *Approximating $\bar{\mathcal{P}}$ with a distribution of form*

$$\bar{\mathcal{Q}} \equiv \hat\pi(X)q(Z|X,\eta)p_\rho(Z'|Z)p(X'|Z',\theta) \quad (7)$$

*where $\hat\pi$ is the empirical data distribution and $q$ and $p$ are the encoder and the decoder models respectively gives the original VAE objective in (3).*
**Proof:** *See Appendix A.1.*

Proposition 2.1 shows that we could have derived the VAE from this extended model as well. This derivation is of course completely redundant since we introduce and cancel out terms. However, we will argue that this extended model is actually more relevant from a representation learning perspective where a coupling $\rho \approx 1$ is preferable as this is the desired behaviour of the encoder.

**Example 2.1. Conditional Generation:** Imagine that we generate a latent $z \sim p(Z)$ and an observation $x \sim p(X|Z = z; \theta)$ from the decoder of a VAE, but consequently discard $z$. Clearly, $x$ is a sample from the marginal $p(X; \theta)$. Now suppose we are told to generate a new sample using the same $z$ as $x' \sim p(X'|Z' = z; \theta)$. As we have discarded $z$, our best bet would be using the extended model with $\rho = 1$ and sample from $\bar{\mathcal{P}}(X'|X = x; \theta, \rho = 1)$ instead.

**Example 2.2. Representation Learning:** Imagine as in the previous example, that we generate a latent $z$ and the corresponding observation $x$, but discard $z$. Suppose we are told to solve a classification task based on a classifier $p(y|z)$. As we have discarded $z$, we would ideally compute an expectation under the true posterior $\int p(y|Z)\bar{\mathcal{P}}(Z|X = x; \eta)dZ$. If our goal is learning the representation, and the task is unknown at training time, we wish to maintain as much as information about $x$. One strategy is asking a faithful reconstruction $x'$ given $x$, i.e., we would like to be able to sample from $\bar{\mathcal{P}}(X'|X = x; \theta, \rho = 1)$ so the goal is not very different from conditional generation.

In practice, $\bar{\mathcal{P}}(X'|X = x; \theta, \rho)$ is not available and we would be using the approximate transition kernel $\bar{\mathcal{Q}}(X'|X; \cdot) \equiv \int \bar{\mathcal{Q}} dZ dZ' / \hat{\pi}(X)$ obtained from the variational distribution. A natural question is how good the approximation $\bar{\mathcal{P}}(X'|X = x; \cdot) \approx \bar{\mathcal{Q}}(X'|X; \cdot)$ is for different $\rho$, and in particular $\rho \approx 1$. Therefore, we will investigate first some properties of this conditional distribution.

**Proposition 2.2.** *The marginal $\bar{\mathcal{P}}(X', X; \theta, \rho)$ is symmetric in $X'$ and $X$, and its marginal does not depend on $\rho$, i.e., $\bar{\mathcal{P}}(X = x; \theta, \rho) = p(X; \theta)$ for any $|\rho| \leq 1$.*
**Proof:** *See Appendix A.2.*

The next proposition shows that if the encoder $q$ is equal to the exact posterior, then $\bar{\mathcal{Q}}(X'|X; \cdot)$ is also exact

**Proposition 2.3.** *If the encoder $q$ and decoder $p$ satisfy the **consistency condition***

$$q(Z|X, \eta)p(X; \theta) = p(X|Z; \theta)p(Z) \tag{8}$$

*then, for all $|\rho| \leq 1$, we have $\bar{\mathcal{Q}}(X'|X; \cdot)p(X; \theta) = \bar{\mathcal{P}}(X', X; \theta, \rho)$. We will say that $\bar{\mathcal{Q}}(X'|X; \cdot)$ is $p(X; \theta)$-invariant.*
**Proof:** *See Appendix A.3.*

The subtle point of Proposition 2.3 is that the exact posterior is valid *for any* coupling strength parameter $\rho$. However, we have seen that the approximation computed by the VAE would be completely agnostic to our choice of $\rho$. Moreover, note that even if the original VAE objective is globally minimized to get $\mathcal{KL}(\hat{\pi}(X)q(Z|X, \eta)||p(X|Z, \theta)p(Z)) = 0$, this may not be sufficient to make sure that the transition kernel $\bar{\mathcal{Q}}(X'|X; \cdot)$ to be $p(X; \theta)$-invariant. The empirical distribution $\hat{\pi}$ has a discrete support and we have no control over the encoder out of this support. Intuitively, we need to introduce additional terms to the objective to steer the encoder if we would like to use the model as a conditional generator, or as a representation especially in the regime $\rho \approx 1$. For this purpose, we will need to make our approximating distribution to match the transition $\bar{\mathcal{P}}(Z'|Z; \cdot) = p_\rho(Z'|Z)$ that is by construction $p(Z)$ invariant.

In Figure 1, we compare what can happen when a VAE is learned nominally with an example of expected behaviour. Here, we show a sequence of images generated by iteratively sampling from a nominally learned encoder-decoder pair on colorMNIST. The drift in the generated images points to a potential inconsistency between the decoder and the encoder. For further insight, we also discuss the special case of probabilistic PCA (Principal Component Analysis) in the appendix Section B, to give an analytically tractable example.

# 3 Autoencoding Variational Autoencoder (AVAE)

Motivated by our analysis, we propose using the following extended model as a target distribution,

$$\mathcal{P}_\rho = p(X|Z; \theta)p(Z)p_\rho(Z'|Z)u(\tilde{X}) \tag{9}$$

where $\rho$ is considered as a fixed hyper-parameter, not to be learned from data. Here $\tilde{X}$ is an additional auxiliary observation (a delusion) that we introduce. We want the target model to be agnostic to its value, hence we choose a flat distribution $u(\tilde{X}) = 1$. We choose as the approximating distribution

$$\mathcal{Q}_{\text{AVAE}} = q(Z'|\tilde{X}; \eta)p_\theta(\tilde{X}|Z)q(Z|X; \eta)\pi(X)$$

The central idea of AVAE is making the encoder and the decoder to be consistent both on the training data and on the auxiliary observations generated by the decoder. We use the notation $p_\theta$ to highlight that the decoder is considered as constant when used as a factor of $\mathcal{Q}_{\text{AVAE}}$ otherwise the original VAE bound would be invalid. Intuitively, the self generated delusion $\tilde{X}$ should not be changing the true log likelihood as otherwise we would be modifying the original objective. The following proposition justifies our choice:

**Proposition 3.1.** *Assume that the consistency condition (8) is true, Then, the transition kernel defined as*

$$\mathcal{Q}_{\text{AVAE}}(Z'|Z; \eta, \theta) \equiv \int q(Z'|\tilde{X}, \eta)p(\tilde{X}|Z; \theta)d\tilde{X}$$

*is $p(Z)$ invariant. Moreover, assume that the latent space has a lower dimension than the observation space ( $Z \in \mathbb{R}^{N_z}$ and $X \in \mathbb{R}^{N_x}$, and $N_z < N_x$), and the decoder mean mapping $g : \mathbb{R}^{N_z} \to \mathbb{X}_g$,*

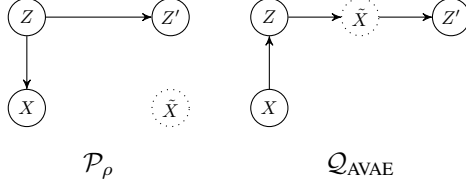

Figure 2: Graphical model of the extended target distribution $\mathcal{P}_\rho$, and the variational approximation $\mathcal{Q}_{\text{AVAE}}$. Here $\tilde{X}$ is a sample generated by the decoder that is subsequently encoded by the encoder.

$\mathcal{P}_\rho$  $\quad\quad\quad\quad$  $\mathcal{Q}_{\text{AVAE}}$

is one-to-one, where $\mathbb{X}_g \subset \mathbb{R}^{N_x}$ is the image of $g$. Then in the limit when the observation noise variance $v$ goes to zero ($v \to 0$) we have $\mathcal{Q}_{AVAE}(Z'|Z; \eta, \theta) = p(Z'|Z; \rho = 1)$.
**Proof:** *See Appendix A.4*

The proposition shows that our choice of the variational approximation (10) is natural: it forces $\mathcal{Q}_{\text{AVAE}}(Z', Z)$ to be close to the marginal of the exact posterior $\bar{\mathcal{P}}(Z', Z) = p_\rho(Z'|Z)p(Z)$. The choice $\mathcal{Q}_{\text{AVAE}}$ is also convenient because it uses the original encoder and decoder as building blocks.

In the appendix Section C.1, we derive the variational objective $\mathcal{B}_{\text{AVAE}} = -\mathcal{KL}(\mathcal{Q}_{\text{AVAE}}||\mathcal{P}_\rho)$. The result is [1]

$$\mathcal{B}_{\text{AVAE}} \;=^+\; -\mathcal{KL}(\pi(X)q(Z|X;\eta)||p(X|Z;\theta)p(Z))$$
$$+ \langle \log p_\rho(Z'|Z) \rangle_{\tilde{q}(Z;\eta)\tilde{q}_\theta(Z'|Z;\eta)} - \left\langle \log q(Z'|\tilde{X};\eta) \right\rangle_{q(Z'|\tilde{X};\eta)\tilde{q}_\theta(\tilde{X};\eta)} \quad (10)$$

where $\tilde{q}(Z;\eta) \equiv \int q(Z|X;\eta)\pi(X)dX$, $\tilde{q}_\theta(\tilde{X};\eta) \equiv \int p_\theta(\tilde{X}|Z)\tilde{q}(Z;\eta)dZ$ and $\tilde{q}_\theta(Z'|Z;\eta) \equiv \int q(Z'|\tilde{X};\eta)p_\theta(\tilde{X}|Z)d\tilde{X}$. In the appendix 1, we provide pseudocode with a stop-gradients primitive to avoid using contributions of $\theta$ dependent terms of $\tilde{q}_\theta$ to the gradient computation.

The resulting objective (10) is intuitive. The first term is identical to the standard VAE ELBO. The second term is a smoothness term that measures the distance between $z$ (the representation that is used to generate the delusion) and $z'$ (the encoding of the delusion). The third term is an extra entropy term on auxiliary observations.

### 3.1 Illustration

As a illustration, we show results for a discrete VAE, where both the encoder and the decoder can be represented as parametrized probability tables. Our goal in choosing a discrete model is visualization of the behaviour of the algorithms in a way that is independent from a particular neural network architecture. The details of this model are explained in the appendix Section D.

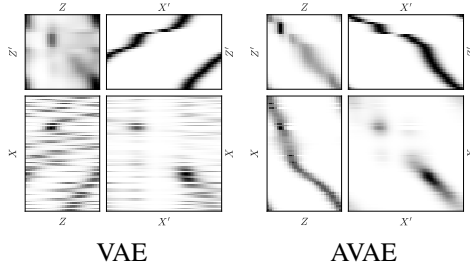

VAE $\quad\quad\quad\quad$ AVAE

Figure 3: Each Panel is showing (from north east clockwise order) heatmaps of probabilities (darker is higher) i) the decoder $p(X'|Z';\theta)$; ii) $\mathcal{Q}(X'|X;\theta,\eta)p(X;\theta)$; iii) the encoder $q(Z|X;\eta)$; iv) $\mathcal{Q}(Z|Z';\theta,\eta)p(Z')$. See text for definitions.

We can visually compare the learned transition models of the learned encoders and decoders for VAE and AVAE in Figure 3, where we show for each model (starting from north east clockwise order) i) the estimated decoder $p(X'|Z';\theta)$ (that is equal in distribution to $p(X|Z;\theta)$ due to parameter tying), ii) the joint distribution $\mathcal{Q}(X'|X;\theta,\eta)p(X;\theta)$, that should be ideally symmetric, where

$$\mathcal{Q}(X'|X;\theta,\eta) = \sum_z p(X'|Z'=z;\theta)q(Z=z|X;\eta)$$

with iii) the encoder $q(Z|X;\eta)$ and iv) the joint distribution $\mathcal{Q}(Z|Z';\theta,\eta)p(Z')$ that should be also ideally symmetric and additionally should be close to identity where

$$\mathcal{Q}(Z|Z';\theta,\eta) = \sum_x q(Z|X=x;\eta)p(X'=x|Z';\theta).$$

We see in Figure 3 that the joint distribution $\mathcal{Q}(Z|Z';\theta,\eta)p(Z')$ is far from an identity mapping, and it is also not symmetrical. In contrast, the distributions learned by AVAE are enforced to be symmetrical, and the joint distribution is concentrated on the diagonal.

**Intermediate summary:** Learning a VAE from data is an highly ill-posed problem and regularization is necessary. Here, instead of modifying the decoder model, we have proposed to constrain the encoder in such a way that it approaches the desired properties of the exact posterior. Our argument started with the extended model in proposition 2.1 that admits the VAE as a marginal. In 2.2, we show that the exact decoder model is by construction independent of the choice of the coupling strength $\rho$. In plain language this means that, if we had access to the exact decoder and if we were able to do exact inference, any $\rho$ will be acceptable. In this ideal case, the extended model would be actually redundant, and in 2.3, we highlight the properties of an exact encoder.

In practice however, we will be learning the decoder from data while doing only approximate amortized inference. Naturally, we want to still retain the properties of the exact posterior. We argue that the extended model is relevant for representation learning (see also example 2.2) and incorporate this explicitly to the encoder by AVAE, and in 3.1 we provide the justification that our choice coincides with the exact target conditional for $\rho = 1$, the representation learning case, when we encode and decode consistently.

We argue that encoder/decoder consistency is related to robustness. The existence of certain 'surprising' adversarial examples [17], where an input image is classified as a very different class after slightly changing input pixels, can be attributed non-smoothness of the representation such as having a large Lipschitz constant, see [5]. The smooth encoder (SE) method proposed in [5] attempted to fix this by data augmentation while training the encoder. In this paper, we introduce a more general framework, (where SE is also a special case) and investigate methods that circumvent the need for computationally costly adversarial attacks during training. The data augmentation is achieved by using the learned generative model itself, as a factor of the approximating $\mathcal{Q}_{\text{AVAE}}$ distribution. The AVAE objective ensures that samples that can be generated by the decoder in the vicinity of the representations corresponding to the training inputs are consistently encoded. In the next experimental section, we will show that the consistency translates to a nontrivial adversarial robustness performance. While our approach does not provide formal guarantees, learning an encoder that retains properties of an exact posterior seems to be central in achieving adversarial robustness.

## 4    Experimental Results

In this section, we will experimentally explore consequences of training with the AVAE objective in (10), implemented as Algorithm 1 in Appendix C. Optimizing the AVAE objective will change the learned encoder and decoder, and we expect that the additional terms will enforce a smooth representation leading to input perturbation robustness in downstream tasks. To test this claim, we will evaluate the encoder in terms of adversarial robustness, using an approach that will be described in the evaluation protocol. To see the effect of the new objective on the decoder and reconstruction quality, we will report Frechet Inception Distance (FID) [9], as well as the test mean square error (MSE).

**Models:** AVAE will be compared to two models, i) VAE trained using the standard ELBO (2), and, ii) Smooth Encoder (SE), a method recently proposed by [5] that uses the same target model as in (6), but uses a different variational approximation. This approximation is computed using adversarial attacks to generate the auxiliary observations $\tilde{X}$. We provide a self contained derivation of this algorithm in Appendix C.2. We also include two hybrid models in our simulations. iii) AVAE-SS (Self Supervised) a post-training method, where a VAE is first trained normally and then post-trained only by samples generated from the decoder (without training data, see Figure 4 for the graphical model – for details see appendix C.4). Finally, we also provide results for a model iv) SE-AVAE, a model that combines both the SE and the AVAE objectives and is concurrently trained using both adversarial attacks and self generated auxiliary observations (for details see appendix C.5). In all models, we use a latent space dimension of 64.

**Data and Tasks:** Experiments are conducted on datasets colorMNIST and CelebA, using both MLP and Convnet architectures in the former, and only Convnet in the latter (for details see Appendix F). The colorMNIST dataset has 2 separate downstream classification tasks (color, digit). For CelebA, we have 17 different binary classification tasks, such as ('has mustache?' or 'wearing hat?').

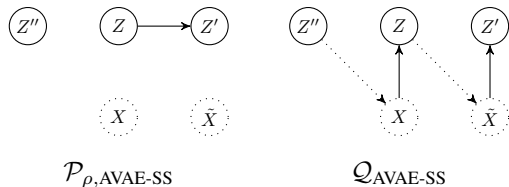

Figure 4: Graphical model of the AVAE-SS target distribution $\mathcal{P}_{\rho,\text{AVAE-SS}}$, and the variational approximation $\mathcal{Q}_{\text{AVAE-SS}}$. Here both $X$ and $\tilde{X}$ is a sample generated by the decoder. The decoder factors (dotted arcs) is fixed (as pretrained by normal VAE).

| Task | digit | | | color | | | Time | MSE | FID |
|------|-------|------|------|-------|-------|------|------|------|------|
| $\epsilon$ | 0.0 | 0.1 | 0.2 | 0.0 | 0.1 | 0.2 | | | |
| VAE | 93.8 | 5.8 | 0.0 | 100.0 | 19.9 | 2.0 | $\times 1$ | 1369.2 | 12.44 |
| $\text{SE}_{0.1}^5$ | 94.3 | 89.6 | 1.8 | 100.0 | 100.0 | 21.8 | $\times 4$ | 1372.5 | 13.01 |
| $\text{SE}_{0.2}^5$ | 95.7 | 92.6 | 87.3 | 100.0 | 99.9 | 99.9 | $\times 4$ | 1374.9 | 11.72 |
| AVAE | 97.3 | 88.1 | 54.8 | 100.0 | 99.8 | 87.7 | $\times 1.5$ | 1371.9 | 15.46 |
| $\text{SE}_{0.1}$-AVAE | 97.4 | 93.6 | 24.5 | 100.0 | 100.0 | 60.0 | $\times 4.7$ | 1373.3 | 13.90 |
| $\text{SE}_{0.2}$-AVAE | 97.6 | 94.2 | 79.8 | 100.0 | 100.0 | 83.2 | $\times 4.7$ | 1374.3 | 13.89 |
| AVAE SS | 94.1 | 72.8 | 20.8 | 100.0 | 99.6 | 56.8 | $\times 1.5$ | 1379.3 | 12.44 |

Table 1: Adversarial test accuracy (in percentage) of the representations for digit and color classification tasks on color MNIST (ConvNet). Evaluation attack radius is $\epsilon$ where pixels are normalized between $[0,1]$. Time is the ratio of the wall-clock time of method to the time taken by the VAE. We show performance of the decoder in terms of MSE and FID score. For AVAE and SE, the coupling strengths are $\rho = 0.975$ and $\rho_{\text{SE}} = 0.95$. The subscript of $\text{SE}_{\epsilon'}$ is the radius $\epsilon'$ used during adversarial training of SE.

**Evaluation Protocol:** The learned representations will be evaluated by their robustness against adversarial perturbations. For the evaluation of adversarial accuracy we employ a three step process, i) first we train an encoder-decoder pair agnostic to any classification task, and ii) subsequently, we freeze the encoder parameters and use the mean mapping $f^\mu$ as a representation to train a linear classifier on top. Thus, each task specific classifier will share the common representation learned by the encoder. This linear classifier is trained normally, *without* any adversarial attacks. Finally, iii) we evaluate the adversarial accuracy of the resulting classifiers. For this, we compute an adversarial perturbation $\delta$ such that $\|\delta\|_\infty \leq \epsilon$ using projected gradient descent (PGD). Here, $\epsilon$ is the attack radius and the optimization goal is changing the classification decision to a different class. The adversarial accuracy is reported in terms of percentage of examples where the attack is not able to find an adversarial example.

**Results:** In Table 1, we show a comparison of the models on colorMNIST, where we evaluate adversarial accuracy for different attack radii $\epsilon$ (0.0 means no attack). Our key observation is that AVAE increases the nominal accuracy and achieves high adversarial robustness, that extends well to strong attacks with a large radius. The surprising fact is that the method is trained completely agnostic to the particular attack type (e.g, PGD), or the attack radius $\epsilon$, and it is able to achieve comparable performance to SE. Due to the computational burden of adversarial attacks, a single iteration of SE takes almost 2.7 times more computation time than a single AVAE iteration, making AVAE practical for large networks. We also observe that for small $\epsilon$, both objectives can be combined to provide improved adversarial accuracy (digit, $\epsilon = 0.1$), however, for higher radius ($\epsilon = 0.2$), the advantage seems to disappear. Another notable algorithm in Table 1 is AVAE-SS, where a pretrained VAE model can be further trained to significantly improve the robustness of the encoder. While this approach seems to be not competitive in terms of the final adversarial accuracy, the fact that it can improve robustness entirely in a self supervised way (Section C.4) is an attractive property.

In Figure 5, we show the summary of several experiments with different architectures, models and various choices of the coupling strength hyper-parameter $\rho$, on the colorMNIST dataset. The first column shows the effect of varying $\rho$, and as expected from our analysis, the adversarial accuracy is high for $\rho$ close to one. The middle column compares different architectures in terms of training dynamics. For MLP, we see that adversarial accuracy slowly increases, while for Convnet, the improvement is very rapid. We also observe that AVAE-SS can significantly improve the robustness of a pretrained VAE, but not to the level of the other methods (results with further $\rho$ are in Appendix G). The third column highlights the behaviour of the algorithms with increasing attack radius. We see

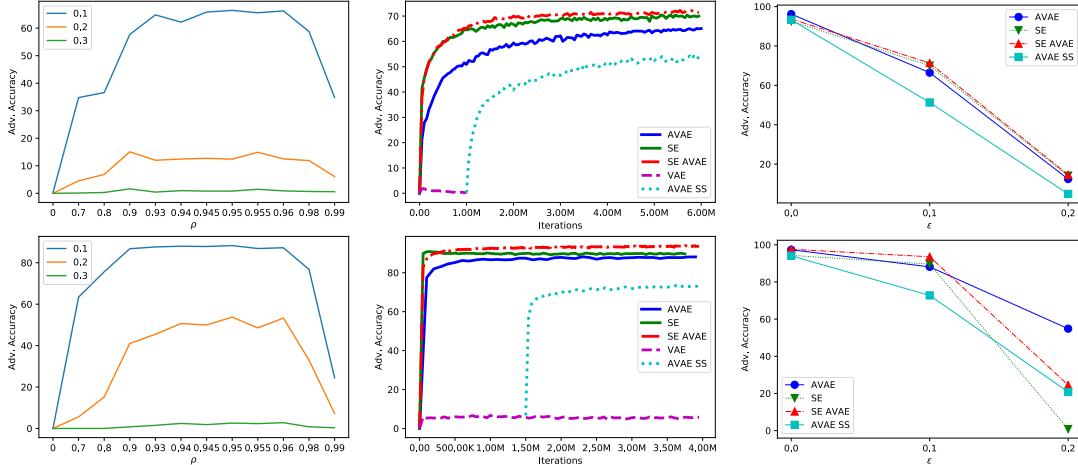

Figure 5: Adversarial accuracy of 'digit' classification task of colorMNIST. (Top row) All results use MLP architecture (Bottom row) Convnet architecture. (Left Column), Adversarial accuracy as a function of the coupling strength $\rho$ for attack radius $\epsilon$. (Middle column) Comparison of algorithms and test adversarial accuracy for attack radius $\epsilon = 0.1$, for $\rho = 0.975$ and $\rho_{SE} = 0.95$. (Right column) Adversarial accuracy as a function of the evaluation attack radius, SE model is trained with $\epsilon' = 0.1$ ($SE_{0.1}$).

| | AVAE | | SE[5] | | SE[20] [5] | | SE[5] AVAE | | AVAE SS | |
|---|---|---|---|---|---|---|---|---|---|---|
| Task / $\epsilon$ | 0.0 | 0.1 | 0.0 | 0.1 | 0.0 | 0.1 | 0.0 | 0.1 | 0.0 | 0.1 |
| Bald | 97.9 | 85.2 | 97.9 | 72.0 | 97.4 | 86.5 | 97.9 | 87.0 | 97.8 | 70.0 |
| Mustache | 96.1 | 91.5 | 95.0 | 69.5 | 95.7 | 84.4 | 96.0 | 92.3 | 94.9 | 74.3 |
| Necklace | 86.1 | 78.4 | 87.8 | 56.7 | 88.0 | 78.9 | 86.1 | 80.3 | 88.0 | 59.7 |
| Eyeglasses | 95.4 | 68.9 | 95.9 | 20.3 | 95.7 | 33.0 | 95.4 | 67.5 | 94.4 | 57.1 |
| Smiling | 77.7 | 3.6 | 87.0 | 3.10 | 85.7 | 1.1 | 77.9 | 6.3 | 81.4 | 0.9 |
| Lipstick | 81.0 | 7.3 | 83.9 | 2.0 | 80.3 | 0.6 | 80.2 | 11.5 | 80.7 | 0.9 |
| Time | $\times 2.2$ | | $\times 3.1$ | | $\times 7.8$ | | $\times 4.3$ | | $\times 2.2$ | |
| MSE | 7276.6 | | 7208.8 | | N/A | | 7269.2 | | 7347.3 | |
| FID | 97.92 | | 98.00 | | N/A | | 109.4 | | 99.8 | |

Table 2: Adversarial test accuracy (in percentage) of the representations for subset of classification tasks on CelebA. For SE methods, superscript $L$ in $SE^L$ denotes the number of PGD iterations used during training of the model.

that for SE, the robustness does not extend beyond the radius that the model was trained for, whereas the accuracy of the AVAE trained model degrades gracefully.

In Table 2, we show that the robustness of a representation learned with AVAE extends to a complex dataset such as CelebA. The VAE results are omitted as they achieve around $0.0$ adversarial accuracy and can be found in Table 3 in Appendix F, along with complete results on all tasks. We see that AVAE performs robustly in downstream tasks, and even surpasses robustness of $SE^5$ and even challenges $SE^{20}$ (reported by [5]) which is substantially requires more computation. We observe that AVAE-SS also achieves non trivial adversarial accuracy across tasks, in some cases even beating $SE^5$. Finally we also report results for SE-AVAE model, showing non-trivial improvements over both SE and AVAE model in most of the downstream tasks. Yet, the increased robustness seems to come with a cost: in all the experiments, we observe an increase in FID (lower is better) and MSE for AVAE models, indicating a tendency of reduced decoder quality and test set reconstruction. We conjecture that this tradeoff may be the result of the encoder being much more constrained than in the VAE case.

# 5    Conclusions

We show that VAE models derived from the canonical formulation in (1) are not unique. We used an alternative model (5) to show that the standard model is unable to capture some desired properties of the exact posterior that are important for representation learning. We proposed a principled alternative, the AVAE, and observed that it gives rise to robust representations without requiring adversarial training. In addition, we present a self supervised variant (AVAE-SS) that also exhibits robustness.

The paper justifies, theoretically and experimentally, the modelling choices of AVAE. Using two benchmark datasets, we demonstrate that the approach can achieve surprisingly high adversarial accuracy without adversarial training. An important feature of the AVAE is that the likelihood function is still equal to the standard VAE likelihood. In doing so, we also provide a principled justification for data augmentation in a density estimation context, where naive data augmentation is not valid as it alters the data distribution. In our framework, the generated data points act like nuisance parameters of the approximating distribution and do not have an effect on the estimated decoder.

Although we are not modifying the original likelihood function, currently, we observe a tradeoff: while the learnt encoder becomes more robust, the corresponding decoder seems to slightly suffer in quality, as measured by FID and MSE. It remains to be seen if there is actually a fundamental reason behind this and we believe that with more carefully designed training methods, both the decoder and the encoder could be in principle improved.

The autoencoding specification is related loosely to the concept of *cycle consistency*, that is often enforced between two data modalities [22]. In [8], authors propose a supervised VAE model for pairs of data points with the same labels to enforce disentanglement by using extra relational information (similar relational models include [18, 13]). In contrast, our approach does only change the encoder. Our method is a VAE based representation learning approach, and there is a large literature on the subject, see, e.g. [20], however models are often not evaluated on their robustness properties. Another fruitful approach in representation learning, especially for the image modality, is contrastive learning [14, 6]. These approaches can challenge and surpass supervised approaches in terms of label efficiency and accuracy, and it remains a future work to investigate the links with our work and whether or not these models can also be used for learning robust representations.

# 6    Societal Impact

**General Research Direction.** In recent years, researchers have trained deep generative models that can generate synthetic examples, often indistinguishable from natural data. The high quality of these samples suggest that these models may be able to learn latent representations useful for other downstream tasks. Learning such representations without task specific supervision facilitates transfer to yet unseen, future tasks. This also fosters label efficiency, and interpretability. Unsupervised representation learning would have a high societal impact as it could enable learning representations from data that can be shared with a wider community of researchers who do not have the computational resources for training such a representations, or do not have direct access to training data due to privacy/security/commercial considerations. However, the properties of these representations in terms of test accuracy, robustness, privacy preservation must be carefully studied before their release, especially if systems will be deployed in the real world. In the current work, we have taken a step towards learning representations in an unsupervised way, that exhibit robustness against transformations of the input.

**Ethical Considerations.** The current work studies representations learned by a specific generative model, the VAE and shares a finding that training a VAE by enforcing an additional natural autoencoding specification is able to provide significant robustness on the learned representation without adversarial training. The study is not proposing a particular system for a specific application. The human face dataset CelebA is choosen as a standard benchmark dataset with several attributes to illustrate the viability of the approach. Still, we have decided to exclude potentially sensitive and subjective attributes from the original dataset, such as 'big-nose' or 'Asian', and use only 17 neutral attributes that we have selected using our own judgment.

## Acknowledgments and Disclosure of Funding

We would like to thank Andriy Minh for the fruitful discussions and excellent feedback.

## Footnotes

[1]When $a$ and $b$ are proportional, we write $\log a =^+ \log b$.

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
