[Supplementary Material]

# A  Proofs

In this section, we provide the proofs of the propositions stated in the main text.

## A.1  Proposition 2.1

*Proof.* By definition

$$\mathcal{KL}(\bar{\mathcal{Q}}|\bar{\mathcal{P}}) = -\left\langle \log \frac{p(X|Z;\theta)p(Z)p_\rho(Z'|Z)p(X'|Z';\theta)}{\hat{\pi}(X)q(Z|X,\eta)p_\rho(Z'|Z)p(X'|Z',\theta)} \right\rangle_{\bar{\mathcal{Q}}} = \mathcal{KL}(\mathcal{Q}|\mathcal{P})$$

Equality follows as for any test function of form $f(X,Z)$, that does not depend on $X'$ and $Z'$ we have $\langle f(X,Z)\rangle_{\bar{\mathcal{Q}}} = \langle f(X,Z)\rangle_{\mathcal{Q}}$. $\qquad\square$

## A.2  Proposition 2.2

*Proof.* $\bar{\mathcal{P}}$ is symmetric (exchangeable) in $(X',Z')$ and $(X,Z)$. Hence, the pairwise marginal

$$\bar{\mathcal{P}}(X',X;\theta,\rho) = \int \bar{\mathcal{P}}(X,X',Z,Z';\theta,\rho)dZdZ' \tag{11}$$

is also symmetric by the parametrization, and $X'$ and $X$ are exchangeable with identical marginal densities. We have $\bar{\mathcal{P}}(X;\theta,\rho) = p(X;\theta)$ as the marginals of $p_\rho(Z',Z)$ do not depend on $\rho$. Hence we have $\bar{\mathcal{P}}(X'|X=x;\theta,\rho) = \bar{\mathcal{P}}(X',X;\theta,\rho)/p(X;\theta)$. $\qquad\square$

## A.3  Proposition 2.3

*Proof.* Consider the joint distribution

$$\bar{\mathcal{Q}}(X'|X;\eta,\theta,\rho)p(X;\theta) = \int \left(q(Z|X,\eta)p(X;\theta)\right)p_\rho(Z'|Z)p(X'|Z',\theta)dZdZ'$$

$$= \int p(X|Z;\theta)p(Z)p_\rho(Z'|Z)p(X'|Z',\theta)dZdZ' = \bar{\mathcal{P}}(X,X';\theta,\rho)$$

The invariance of $p(X;\theta)$ follows from symmetry as $\int \bar{\mathcal{Q}}(X'|X;\cdot)p(X;\theta) = p(X';\theta)$. $\qquad\square$

## A.4  Proposition 3.1

*Proof.* Consider

$$\mathcal{Q}_{\text{AVAE}}(Z'|Z;\eta,\theta)p(Z) \equiv \int q(Z'|\tilde{X},\eta)p(\tilde{X}|Z;\theta)p(Z)d\tilde{X} = \int q(Z'|\tilde{X},\eta)q(Z|\tilde{X},\eta)p(\tilde{X};\theta)d\tilde{X}$$

As the expression at the right is symmetric in $Z$ and $Z'$, $p(Z)$-invariance follows. When the observation noise $v \to 0$, the probability of observing $\tilde{x}$ out of the image of $g$ vanishes, i.e., $\Pr\{\tilde{x} \notin \mathbb{X}_g\} = 0$. When $g$ is continuous and differentiable, the image of $g$ is a manifold $\mathbb{X}_g$. As $g$ is one-to-one, it is invertible on $\mathbb{X}_g$, $\tilde{x} = g(z;\theta) \Leftrightarrow g^{-1}(\tilde{x}) = z$ and the optimal encoder will have the mean mapping $f^\mu(\tilde{x};\eta^*) = g^{-1}(\tilde{x};\eta^*)$ and variance mapping $f^\Sigma(\tilde{x};\eta^*) = 0$. As $z' = g^{-1}(\tilde{x})$ and $\tilde{x} = g(z)$, we have $z' = z$ and have $\mathcal{Q}_{\text{AVAE}}(Z'|Z;\eta,\theta) = p(Z'|Z;\rho=1)$. $\qquad\square$

# B  Example: PCA case

As a special case, it is informative to consider the probabilistic principal component analysis (pPCA) [16, 19], a special case of VAE where $g$ and $f^\mu$ are constrained to be linear functions. When $g(Z;\theta=W) = Wz$, using standard results about Gaussian distributions, the optimal encoder is given by the exact posterior and is available in closed form as

$$q_*(Z|X=x) = \mathcal{N}(Z; f_*^\mu(x), f_*^\Sigma)$$

where $f_*^\mu(x) = (W^\top W + vI)^{-1}W^\top x$, $f_*^\Sigma = v(W^\top W + vI)^{-1}$. The transition kernel can be shown to be the following form

$$\mathcal{Q}(X'|X=x) = \mathcal{N}(X'; P_{W,v}x, v(P_{W,v}+I))$$

where $P_{W,v} = W(W^\top W + vI)^{-1}W^\top$.

In the limit when $v$ is zero, we have the PCA case and the mean mapping of the transition kernel $g \circ f_*^\mu$ corresponds to a projection matrix $P_{W,0} = P_W = W(W^\top W)^{-1}W^\top$, as $P_W = P_W^2$. Hence all noise will vanish, and iterating the encode-decode steps would generate the sequence $P_W x_0 = x_1 = x_2 = \ldots$; any initial input will be projected first into the range space of the projector, and subsequent points will be confined to the invariant subspace.

Note that the encoder and the decoder are also consistent, as any possible sample $x$ that can be generated by the decoder, is mapped to a representation in the vicinity of the original representation. To see this, consider the transition kernel that can be shown to be

$$\mathcal{Q}(Z'|Z = z) \quad = \quad \mathcal{N}(Z'; J_{W,v}z, S)$$

where $J_{W,v} = (W^\top W + vI)^{-1}W^\top W$ and $S = v(J_{W,v} + I)(W^\top W + vI)^{-1}$. In the limit, when $v$ goes to zero, we have $J_{W,v} \approx I$, or equivalently $z' = z$.

However, if an 'inconsistent' decoder-encoder pair would be used, an encoder with a perturbed mean transform $(W^\top W + vI)^{-1}W^\top + \Delta$ for some nonzero matrix $\Delta$, the resulting transition matrix of $\mathcal{Q}(X'|X = x)$ won't be in general a projection and the chains will 'drift away' from the original invariant subspace depending upon the norm of the perturbation and the spectrum of the resulting matrix $W\Delta$.

In the PCA case, the invariant subspace is explicitly known thanks to the linearity. In contrast, for a VAE with flexible nonlinear decoder and encoder functions, the analogous object to an invariant subspace would be a global invariant manifold (sometimes referred as the data manifold), embedded in $\mathcal{X}$ and each manifold point charted by $x = g(z; \theta)$ where $z$ is the coordinate of $x$ in $\mathcal{Z}$, given by $f^\mu(x)$. While it seems to be hard to characterize the invariant manifold, we can argue that "autoencoding" requires that realizations generated by the decoder are approximately invariant when encoded again; i.e. autoencoding means that the coordinate of the point $g(z)$ is $z$, or equivalently $z \approx f^\mu(g(z))$. However, this requirement is enforced by the VAE only for the samples in the dataset but not for typical samples $z$ from the prior $p(Z)$. So the name autoencoder is perhaps a misnomer as the resulting model is not necessarily autoencoding the samples it can generate.

## C  Variational Approximations

In this appendix, we provide the derivation of the variational objective for the AVAE and then an alternative derivation for the smooth encoder [5], and other hybrid models that we used in our experiments as contender models.

### C.1  Autoencoding Variational Autoencoder $\mathcal{Q}_{\text{AVAE}}$

The AVAE objective is

$$(\theta^*, \eta^*) = \arg\max_{\theta,\eta} \mathcal{B}_{\text{AVAE}}(\theta, \eta)$$

where

$$\mathcal{B}_{\text{AVAE}}(\theta, \eta) \quad = \quad -\mathcal{KL}(\mathcal{Q}_{\text{AVAE}}|\mathcal{P}_\rho) = \langle \log(\mathcal{P}_\rho/\mathcal{Q}_{\text{AVAE}}) \rangle_{\mathcal{Q}_{\text{AVAE}}}$$

The target distribution is

$$\mathcal{P}_\rho = p(X|Z; \theta)p(Z)p_\rho(Z'|Z)u(\tilde{X}) \tag{12}$$

where $\rho$ is considered as a fixed hyper-parameter and $u(\tilde{x}) = 1$. The approximating distribution is

$$\mathcal{Q}_{\text{AVAE}} \quad = \quad \pi(X)q(Z|X; \eta)q(Z'|\tilde{X}; \eta)p_\theta(\tilde{X}|Z)$$

$$\mathcal{B}_{\text{AVAE}}(\theta, \eta) \quad = \quad \left\langle \log \frac{p(X|Z; \theta)p(Z)p_\rho(Z'|Z)u(\tilde{X})}{\pi(X)q(Z|X; \eta)q(Z'|\tilde{X}; \eta)p_\theta(\tilde{X}|Z)} \right\rangle_{\mathcal{Q}_{\text{AVAE}}}$$

$$= \quad -\mathcal{KL}(\pi(X)q(Z|X; \eta)|p(X|Z; \theta)p(Z))$$

$$+ \left\langle \log \frac{p_\rho(Z'|Z)}{q(Z'|\tilde{X}; \eta)p_\theta(\tilde{X}|Z)} \right\rangle_{\tilde{q}(Z;\eta)q(Z'|\tilde{X};\eta)p_\theta(\tilde{X}|Z)}$$

We have used $p_\theta$ in the subscript to denote the fact that the decoder parameters will be assumed to be fixed. Cancelling the constant terms we obtain

$$
\begin{aligned}
\mathcal{B}_{\text{AVAE}} \;=^{+}\; & \langle \log p(X|Z;\theta)\rangle_{\pi(X)q(Z|X;\eta)} + \langle \log p(Z)\rangle_{\pi(X)q(Z|X;\eta)} - \langle \log q(Z|X;\eta)\rangle_{\pi(X)q(Z|X;\eta)} \\
& + \langle \log p(Z'|Z;\rho)\rangle_{\tilde{q}(Z;\eta)\tilde{q}(Z'|Z;\eta)} - \left\langle \log q(Z'|\tilde{X};\eta)\right\rangle_{q(Z'|\tilde{X};\eta)\tilde{q}(\tilde{X};\eta)}
\end{aligned}
$$

where $\tilde{q}_\theta(Z'|Z;\eta) \equiv \int q(Z'|\tilde{X};\eta)p_\theta(\tilde{X}|Z)d\tilde{X}$, $\tilde{q}(\tilde{X};\eta) \equiv \int q(\tilde{X}|X;\eta)\pi(X)dX$ and $\tilde{q}(Z;\eta) \equiv \int q(Z|X;\eta)\pi(X)dX$.

The algorithm is shown in Algorithm 1. We approximate the required expectations by their Monte Carlo estimates and the objective function to maximize becomes

$$
\begin{aligned}
\hat{\mathcal{B}}_{\text{AVAE}}(\theta,\eta;x,\tilde{x},z) \;=\; & \log p(X=x|Z=z;\theta) \\
& + \langle \log p(Z)\rangle_{q(Z|X=x;\eta)} + \langle \log p(Z'|Z;\rho)\rangle_{\tilde{q}(Z|X=x;\eta)\tilde{q}(Z'|\tilde{X}=\tilde{x};\eta)} \\
& - \left\langle \log q(Z'|\tilde{X}=\tilde{x};\eta)\right\rangle_{q(Z'|\tilde{X}=\tilde{x};\eta)} \\
& - \langle \log q(Z|X=x;\eta)\rangle_{q(Z|X=x;\eta)}
\end{aligned} \tag{13}
$$

---

**Algorithm 1: AVAE Training**

1: **function** TRAINAVAE($x$, iterations)
2:      encoder $\leftarrow (f^\mu(\cdot;\eta), f^\Sigma(\cdot;\eta))$
3:      decoder $\leftarrow g(\cdot;\theta)$
4:      **for** $i \leftarrow 1$ to iterations **do**
5:          $\mu_x, \Sigma_x^{1/2} \leftarrow$ encoder($x$)
6:          $z \leftarrow \mu_x + \Sigma_x^{1/2}\mathcal{N}(0,1)$
7:          $\tilde{x} \leftarrow stopgradient(\text{decoder}(z))$
8:          $\mu_{\tilde{x}}, \Sigma_{\tilde{x}}^{1/2} \leftarrow$ encoder($\tilde{x}$)

9:          Maximize$_{\theta,\eta}$ $\hat{\mathcal{B}}_{\text{AVAE}}(\theta,\eta;x,\tilde{x},z)$
            $\triangleright$ See (13)
10:      **end for**
11: **end function**

---

**Algorithm 2: SE Training**

1: **function** TRAINSE($x$, iterations)
2:      encoder $\leftarrow (f^\mu(\cdot;\eta), f^\Sigma(\cdot;\eta))$
3:      decoder $\leftarrow g(\cdot;\theta)$
4:      **for** $i \leftarrow 1$ to iterations **do**
5:          $\tilde{x} \leftarrow stopgradient(\text{PGD}(x))$
6:          $\mu_x, \Sigma_x^{1/2} \leftarrow$ encoder($x$)
7:          $z \leftarrow \mu_x + \Sigma_x^{1/2}\mathcal{N}(0,1)$

8:          Maximize$_{\theta,\eta}$ $\hat{\mathcal{B}}_{\text{SE}}(\theta,\eta;x,\tilde{x},z)$
            $\triangleright$ See (32)
9:      **end for**
10: **end function**

---

**Algorithm 3: SE-AVAE Training**

1: **function** TRAINSE-AVAE($x$, iterations)
2:      encoder $\leftarrow (f^\mu(\cdot;\eta), f^\Sigma(\cdot;\eta))$
3:      decoder $\leftarrow g(\cdot;\theta)$
4:      **for** $i \leftarrow 1$ to iterations **do**
5:          $\tilde{\tilde{x}} \leftarrow stopgradient(\text{PGD}(x))$
6:          $\mu_x, \Sigma_x^{1/2} \leftarrow$ encoder($x$)
7:          $z \leftarrow \mu_x + \Sigma_x^{1/2}\mathcal{N}(0,1)$
8:          $\tilde{x} \leftarrow stopgradient(\text{decoder}(z))$

9:          Maximize$_{\theta,\eta}$
     $\hat{\mathcal{B}}_{\text{SE-AVAE}}(\theta,\eta;x,\tilde{x},\tilde{\tilde{x}},z)$
            $\triangleright$ See (35)
10:      **end for**
11: **end function**

---

**Algorithm 4: AVAE-SS Training**

1: **function** TRAINAVAE-SS($x$, iterations)
2:      encoder $\leftarrow (f^\mu(\cdot;\eta), f^\Sigma(\cdot;\eta))$
3:      decoder $\leftarrow g(\cdot;\theta)$
4:      **for** $i \leftarrow 1$ to iterations **do**
5:          $z'' \leftarrow \mathcal{N}(0,1)$
6:          $x \leftarrow stopgradient(\text{decoder}(z''))$
7:          $\mu_x, \Sigma_x^{1/2} \leftarrow$ encoder($x$)
8:          $z \leftarrow \mu_x + \Sigma_x^{1/2}\mathcal{N}(0,1)$
9:          $\tilde{x} \leftarrow stopgradient(\text{decoder}(z))$

10:     Maximize$_{\theta,\eta}$ $\hat{\mathcal{B}}_{\text{AVAE-SS}}(\theta,\eta;x,\tilde{x},z)$
            $\triangleright$ See (34)
11:     **end for**
12: **end function**

---

## C.2 Smooth Encoder

Smooth encoder model proposed in [5] employs an alternative variational approximation strategy to learn a representation. While SE introduced an 'external selection mechanism' to generate adversarial examples, the analysis in this appendix shows that the approach could be viewed as a robust Bayesian approach to variational inference and choosing a different variational distribution than AVAE.

SE aims at learning a model that is insensitive to a class of input transformations, in particular small input perturbations $T_\alpha(x) = x + \alpha$ where $\alpha \in A = \{\alpha : \|\alpha\| \leq \epsilon\}$. The variational distribution has the form

$$\mathcal{Q}_{\text{SE}} = \pi(X)q(Z|X;\eta)q_T(\tilde{X}|X;u_A)q(Z'|\tilde{X};\eta) \tag{14}$$

where $q_T(\tilde{X}|X;u_A)$ is the conditional distribution defined by

$$\alpha \sim u_A(\alpha) \qquad \tilde{X} = T_\alpha(X) \tag{15}$$

Here, $u_A$ is an arbitrary distribution with $u_A \in \mathcal{U}_A$, where $\mathcal{U}_A$ is the set of distributions defined on $A$. The notation suggests that $u_A$ is now taken as a parameter. The bound is

$$\tilde{\mathcal{B}}_{\text{SE}}(\eta, \theta; u_A) = -\mathcal{KL}(\mathcal{Q}_{\text{SE}}(\eta, u_A)||\mathcal{P}_\rho(\theta)) \tag{16}$$

We can employ a robust Bayesian approach to define a 'pessimistic' bound in the sense of selecting the worst prior distribution $u_A$

$$\mathcal{B}_{\text{SE}}(\eta, \theta) = \min_{u_A \in \mathcal{U}_A} \tilde{\mathcal{B}}_{\text{SE}}(\eta, \theta; u_A) \tag{17}$$

This optimization is still computationally feasible as the maximum will be attained by a degenerate distribution concentrated on an adversarial example $x_a = x + \alpha$ as $u_A(\alpha) = \delta(\alpha - (x_a - x))$ and can be computed using projected gradient descent to find $x_a$. Once $u_A$ is fixed, we can optimize model parameters in the outer maximization.

The resulting algorithm has two steps i) (Augmentation) Generate a new empirical data distribution $\hat{\pi}_a(\tilde{X}|X)$ adversarially, by finding the worst case transformation for each data point in the sense of maximizing the change between representations; and ii) (Maximization) Maximizing the bound denoted by $\mathcal{B}_{\text{SE}}$ that has the following form

$$\mathcal{B}_{\text{SE}} \ =^+ \ \langle \log p(X|Z;\theta) \rangle_{\pi(X)q(Z|X;\eta)} - \mathcal{KL}(\tilde{\pi}_a(X,\tilde{X})q(Z,Z'|X,\tilde{X};\eta)|p(Z,Z';\rho)) \tag{18}$$

where $q(Z, Z'|X, \tilde{X}; \eta) \equiv q(Z|X;\eta)q(Z'|\tilde{X};\eta)$ and $\tilde{\pi}_a(X, \tilde{X}) \equiv \pi(X)\hat{\pi}_a(\tilde{X}|X)$.

This objective measures the data fidelity (first term) and the divergence of the joint encoder mapping from the pairwise coupling target (second term). The second term forces the encoder mapping to be smooth when $\rho \approx 1$.

## C.3 Derivation Details

- The SE objective:

$$(\theta^*, \eta^*) = \arg\max_{\theta, \eta} \min_{u_A \in \mathcal{U}_A} \tilde{\mathcal{B}}_{\text{SE}}(\theta, \eta; u_A)$$

$$\tilde{\mathcal{B}}_{\text{SE}}(\theta, \eta; u_A) \ = \ -\mathcal{KL}(\mathcal{Q}_{\text{SE}}(\eta; u_A)|\mathcal{P}_\rho(\theta)) = \langle \log(\mathcal{P}_\rho/\mathcal{Q}_{\text{SE}}) \rangle_{\mathcal{Q}_{\text{SE}}}$$

- Target:

$$\mathcal{P}_\rho = p(X|Z;\theta)p(Z)p_\rho(Z'|Z)u(\tilde{X}) \tag{19}$$

  fixed hyper-parameter $\rho \in [0, 1)$ and $u(\tilde{x}) = 1$.

- The variational distribution:

$$\mathcal{Q}_{\text{SE}}(\eta; u_A) = \pi(X)q(Z|X;\eta)q_T(\tilde{X}|X;u_A)q(Z'|\tilde{X};\eta) \tag{20}$$

  - $\pi(X)$: Data distribution to be replaced by empirical data distribution
  - $q(Z|X;\eta), q(Z'|\tilde{X};\eta)$: Encoders tied with same parameters $\eta$

– $q_T(\tilde{X}|X; u_A)$: Distribution induced by random translations

$$\alpha \sim u_A(\alpha) \qquad \tilde{X} = X + \alpha \tag{21}$$

Here, $u_A$ is an arbitrary distribution on $A = \{a : \|a\|_\infty \le \epsilon\}$ with $u_A \in \mathcal{U}_A$.

$$
\begin{aligned}
\tilde{\mathcal{B}}_{\text{SE}}(\theta, \eta; u_A) \quad =^+ \quad & \langle \log p(X|Z; \theta) p(Z) p_\rho(Z'|Z) \rangle_{\mathcal{Q}_{\text{SE}}} \\
& - \left\langle \log q(Z|X; \eta) q(Z'|\tilde{X}; \eta) \right\rangle_{\mathcal{Q}_{\text{SE}}} - \left\langle \log q_T(\tilde{X}|X; u_A) \right\rangle_{\mathcal{Q}_{\text{SE}}} \tag{22} \\
= \quad & \langle \log p(X|Z; \theta) \rangle_{\pi(X)q(Z|X;\eta)} \\
& + \langle \log p(Z) \rangle_{\pi(X)q(Z|X;\eta)} \\
& - \langle \log q(Z|X; \eta) \rangle_{\pi(X)q(Z|X;\eta)} \\
& + \langle \log p_\rho(Z'|Z) \rangle_{\pi(X)q(Z|X;\eta)q_T(\tilde{X}|X;u_A)q(Z'|\tilde{X};\eta)} \\
& - \left\langle \log q(Z'|\tilde{X}; \eta) \right\rangle_{\pi(X)q(Z|X;\eta)q_T(\tilde{X}|X;u_A)q(Z'|\tilde{X};\eta)} \\
& - \left\langle \log q_T(\tilde{X}|X; u_A) \right\rangle_{\pi(X)q_T(\tilde{X}|X;u_A)} \tag{23}
\end{aligned}
$$

The objective is

$$(\eta^*, \theta^*) = \arg \max_{\eta, \theta} \min_{u_A \in \mathcal{U}_A} \tilde{\mathcal{B}}_{\text{SE}}(\theta, \eta; u_A) \tag{24}$$

Iterative optimization for $\tau = 1, 2, \dots$:

- Augmentation step: Solve or improve $u_A^{(\tau)} = \arg \min_{u_A \in \mathcal{U}_A} \tilde{\mathcal{B}}_{\text{SE}}(\theta^{(\tau-1)}, \eta^{(\tau-1)}; u_A)$

- Maximization step: Solve or improve $(\eta^{(\tau)}, \theta^{(\tau)}) = \arg \max_{\eta, \theta} \tilde{\mathcal{B}}_{\text{SE}}(\theta, \eta; u_A^{(\tau)})$

**Augmentation step:** In the inner optimization we seek for the worst case $u_A$ that will minimize the ELBO, or equivalently

$$u_A^* = \arg \min_{u_A \in \mathcal{U}_A} \left\{ \tilde{\mathcal{B}}_{\text{SE}}(\theta, \eta; u_A) \right\}$$

While this optimization seems to be on the space of all distributions, we can see that the last term (23) of the bound $\mathcal{B}_{\text{SE}}$ is the entropy of $q_T$, $H[q_T]$. Consequently, the bound $\tilde{\mathcal{B}}_{\text{SE}}$ is minimized when the entropy $H[q_T]$ is minimized, i.e. when $q_T$ is degenerate, and concentrated on a point. Consequently, focusing on the remaining terms we can rewrite this optimization problem for each data point as

$$\alpha^* = \arg \max_{\alpha \in A} \left\{ - \langle \log p(Z'|Z; \rho) \rangle_{q(Z|X=x;\eta)q(Z'|\tilde{X}=T_\alpha(x);\eta)} \right. \tag{25}$$

$$\left. - H[q(Z'|\tilde{X} = T_\alpha(x); \eta)] - H[q(Z|X = x; \eta)] \right\} \tag{26}$$

$$\tilde{x}_a = x + \alpha^* \tag{27}$$

This term can be identified as a lower bound to the entropy regularized $\ell_2$ optimal transport [5] hence is measuring the discrepancy between $q(Z|X = x; \eta)$ and $q(Z'|\tilde{X} = \tilde{x}; \eta)$. It can be interpreted as an adversarial attack trying to maximize the change in the representations. We will denote the empirical distribution that is obtained by attacking each sample adversarially as $\hat{\pi}_a(\tilde{X}|X) = \sum_i \delta(\tilde{X} - \tilde{x}_a^{(i)})$.

**Maximization step:** Given the empirical distribution of inputs and their augmentation via adversarial attacks, denoted as $\hat{\pi}_a(X, \tilde{X}) \equiv \pi(X)\hat{\pi}_a(\tilde{X}|X)$ is fixed in an iteration, the objective is

$$
\begin{aligned}
\tilde{\mathcal{B}}_{\text{SE}}(\theta, \eta) \quad =^+ \quad & \langle \log p(X|Z; \theta) \rangle_{\pi(X)q(Z|X;\eta)} \\
& + \langle \log p(Z) \rangle_{\pi(X)q(Z|X;\eta)} \\
& - \langle \log q(Z|X; \eta) \rangle_{\pi(X)q(Z|X;\eta)} \\
& + \langle \log p_\rho(Z'|Z) \rangle_{q(Z|X;\eta)q(Z'|\tilde{X};\eta)\hat{\pi}_a(X,\tilde{X})} \\
& - \left\langle \log q(Z'|\tilde{X}; \eta) \right\rangle_{q(Z|X;\eta)q(Z'|\tilde{X};\eta)\hat{\pi}_a(X,\tilde{X})} \\
= \quad & \langle \log p(X|Z; \theta) \rangle_{\pi(X)q(Z|X;\eta)} \\
& -\mathcal{KL}(\hat{\pi}_a(X, \tilde{X})q(Z, Z'|X, X'; \eta)|p_\rho(Z', Z)) \qquad (28)
\end{aligned}
$$

where $q(Z, Z'|X, X'; \eta) \equiv q(Z|X; \eta)q(Z'|\tilde{X}; \eta)$.

### C.3.1 A Tighter bound

Even though we derived the above bound using a factorized distribution assumption $q(Z, Z'|X, X'; \eta) = q(Z|X; \eta)q(Z'|\tilde{X}; \eta)$, [5] is using a tighter bound using

$$
q(Z', Z|X = x, \tilde{X} = \tilde{x}; \eta) = \mathcal{N} \left( \begin{pmatrix} f^\mu(x; \eta) \\ f^\mu(\tilde{x}; \eta) \end{pmatrix}, \begin{pmatrix} f^\Sigma(x; \eta) & \psi \\ \psi & f^\Sigma(\tilde{x}; \eta) \end{pmatrix} \right)
$$

Here, $\psi$ is a diagonal matrix with the $i$'th diagonal element

$$
\psi_i \quad = \quad \frac{1}{2\gamma} \left( \sqrt{1 + 4\gamma^2 f^\Sigma(x; \eta)_i f^\Sigma(\tilde{x}; \eta)_i} - 1 \right)
$$

where $\gamma \equiv \rho/(1 - \rho^2)$.

ith this modification we get

$$
\begin{aligned}
\langle \log p_\rho(Z', Z) \rangle_{\tilde{q}(Z', Z|X=x, \tilde{X}=\tilde{X};\eta)} \quad = \quad & -\frac{1}{1 - \rho^2} \left( f^\mu(x; \eta) + f^\Sigma(x; \eta) + f^\mu(\tilde{x}; \eta) + f^\Sigma(\tilde{x}; \eta) \right) \\
& + \frac{2\rho}{1 - \rho^2} \left( \psi + f^\mu(x; \eta)f^\mu(\tilde{x}; \eta) \right) + C \qquad (29)
\end{aligned}
$$

Under the $q$ distribution, the desired expectations are

$$
\left\langle \mathbf{Tr}\, Z'Z'^\top \right\rangle = \mathbf{Tr}(\tilde{\Sigma} + \tilde{\mu}\tilde{\mu}^\top) \quad \left\langle ZZ^\top \right\rangle = \mathbf{Tr}(\Sigma + \mu\mu^\top) \quad \left\langle Z'Z^\top \right\rangle = \mathbf{Tr}(\psi + \tilde{\mu}\mu^\top) \quad (30)
$$

$\mu \equiv f^\mu(x; \eta), \tilde{\mu} \equiv f^\mu(\tilde{x}; \eta), \Sigma \equiv f^\Sigma(x; \eta), \tilde{\Sigma} \equiv f^\Sigma(\tilde{x}; \eta)$

$$
\langle \log p_\rho(Z', Z) \rangle_{\tilde{q}} \quad = \quad -\frac{1}{2(1 - \rho^2)} \mathbf{Tr} \left( \tilde{\Sigma} + \Sigma + \tilde{\mu}\tilde{\mu}^\top + \mu\mu^\top - 2\rho(\psi + \tilde{\mu}\mu^\top) \right) \qquad (31)
$$

With the given tighter bound the algorithm for SE is shown in Algorithm 2. From Equation 18 we approximate the required expectations by their Monte Carlo estimates and the objective function to maximize becomes

$$
\begin{aligned}
\hat{\mathcal{B}}_{\text{SE}}(\theta, \eta; x, \tilde{x}, z) \quad = \quad & \langle \log p(X = x|Z = z) \rangle_{q(Z|X=x;\eta)} \\
& + \langle \log p(Z', Z) \rangle_{\tilde{q}(Z,Z'|X=x, \tilde{X}=\tilde{x})} \\
& - \langle \log \tilde{q}(Z, Z') \rangle_{\tilde{q}(Z,Z'|\tilde{X}=\tilde{x}, X=x)} \qquad (32)
\end{aligned}
$$

Figure 6: Graphical model of the AVAE-SS target distribution $\mathcal{P}_{\rho,\text{AVAE-SS}}$, and the variational approximation $\mathcal{Q}_{\text{AVAE-SS}}$. Here both $X$ and $\tilde{X}$ is a sample generated by the decoder.

$\mathcal{P}_{\rho,\text{AVAE-SS}}$  $\mathcal{Q}_{\text{AVAE-SS}}$

## C.4 AVAE-SS (Self Supervised)

This algorithm can be used for post training an already trained VAE. Figure 6 shows the graphical model describing AVAE-SS model.

$$
\begin{aligned}
\mathcal{B}_{\text{AVAE-SS}} \;=^{+}\; & -\mathcal{KL}\Big(p(Z'')p(X|Z'')q(Z|X;\eta)p(\tilde{X}|Z)q(Z'|\tilde{X})||p(Z'')p(Z'|Z)p(Z)u(\tilde{X})u(X)\Big) \\
= & \;\langle \log p(Z)\rangle_{q(Z|X)} + \langle \log p(Z'|Z)\rangle_{q(Z|X)q(Z'|\tilde{X})} \\
& - \Big\langle \log p(Z'')p(X|Z'')q(Z|X;\eta)p(\tilde{X}|Z)q(Z'|\tilde{X})\Big\rangle_{p(X|Z'')q(Z|X;\eta)p(\tilde{X}|Z)q(Z'|\tilde{X})}
\end{aligned}
\tag{33}
$$

The algorithm is shown in Algorithm 4. We approximate the required expectations by their Monte Carlo estimates and the objective function to maximize becomes

$$
\begin{aligned}
\hat{\mathcal{B}}_{\text{AVAE-SS}}(\theta,\eta;x,\tilde{x},z) \;=\; & \langle \log p(Z)\rangle_{q(Z|X=x;\eta)} + \langle \log p(Z'|Z;\rho)\rangle_{\tilde{q}(Z|X=x;\eta)\tilde{q}(Z'|\tilde{X}=\tilde{x};\eta)} \\
& - \Big\langle \log q(Z'|\tilde{X}=\tilde{x};\eta)\Big\rangle_{q(Z'|\tilde{X}=\tilde{x};\eta)} \\
& - \langle \log q(Z|X=x;\eta)\rangle_{q(Z|X=x;\eta)}
\end{aligned}
\tag{34}
$$

Also see Section C.1 for further expansion on terms.

## C.5 SE-AVAE

Figure 7 shows the graphical model describing AVAE-SS model.

$$
\begin{aligned}
\mathcal{B}_{\text{SE-AVAE}} \;=^{+}\; & -\mathcal{KL}(\pi(X)p(\tilde{\tilde{X}}|X)q(Z''|\tilde{\tilde{X}};\eta)q(Z|X)p(\tilde{X}|Z;\eta)q(Z'|\tilde{X}) \\
& \quad ||p(Z'|Z)p(Z''|Z)p(X|Z)p(Z))
\end{aligned}
$$

The algorithm is shown in Algorithm 3. We approximate the required expectations by their Monte Carlo estimates and the objective function to maximize becomes

$$
\begin{aligned}
\hat{\mathcal{B}}_{\text{SE-AVAE}}(\theta,\eta;x,\tilde{x},\tilde{\tilde{x}},z) \;=\; & \langle \log p(Z)\rangle_{q(Z|X=x;\eta)} + \langle \log p(Z'|Z;\rho)\rangle_{\tilde{q}(Z|X=x;\eta)\tilde{q}(Z'|\tilde{X}=\tilde{x};\eta)} \\
& + \langle \log p(Z''|Z;\rho_{\text{SE}})\rangle_{\tilde{q}(Z|X=x;\eta)\tilde{q}(Z''|\tilde{\tilde{X}}=\tilde{\tilde{x}};\eta)} \\
& - \Big\langle \log q(Z'|\tilde{X}=\tilde{x};\eta)\Big\rangle_{q(Z'|\tilde{X}=\tilde{x};\eta)} \\
& - \langle \log q(Z|X=x;\eta)\rangle_{q(Z|X=x;\eta)} \\
& - \Big\langle \log q(Z''|\tilde{\tilde{X}}=\tilde{\tilde{x}};\eta)\Big\rangle_{q(Z''|\tilde{\tilde{X}}=\tilde{\tilde{x}};\eta)}
\end{aligned}
\tag{35}
$$

# D  Details of the 1-D Example

In this example Section 3.1, we will assume that both the observations $x$ and latents $z$ can only take values from discrete sets and to avoid boundary effects we adopt a parametrization reminiscent of a

Figure 7: Graphical model of the target distribution $\mathcal{P}_{\rho,\rho_{\text{SE}}}$, and the variational approximation $\mathcal{Q}_{\text{SE-AVAE}}$. Here $\tilde{X}$ is a sample generated by the decoder that is subsequently encoded by the encoder.

Figure 8: Results of a VAE. (Left to right) i) The empirical data distribution $\hat{\pi}(X)$, ii) Encoder weighted by the empirical distribution $\hat{\pi}(X)q(Z|X;\eta)$, iii) Encoder weighted by the model distribution $p(X;\theta)q(Z|X;\eta)$, iv) the decoder $p(X|Z)p(Z)$, v) the model distribution $p(X;\theta)$, obtained by marginalizing the decoder.

von-Mises distribution with normalization constant $J$:

$$\mathcal{VM}(X;\mu,v) \equiv \frac{1}{J(\mu,v)}\exp\left(\cos\left(X-\mu\right)/v\right) \qquad J(\mu,v) = \sum_{x\in\mathcal{X}}\exp\left(\cos\left(X-\mu\right)/v\right)$$

As these densities (up to quantization effects) are unimodal and symmetric around their means with a bell-shape, this example is qualitatively similar to a standard conditionally Gaussian VAE. We define the following system of conditional distributions as the decoder and encoder models as:

$$p(X|Z=z;\theta) = \mathcal{VM}(X;g(z;\theta),v) \qquad q(Z|X=x;\eta) = \mathcal{VM}(Z;f^{\mu}(x;\eta),f^{\Sigma}(x;\eta))$$

where we let $X \in \{c_x, 2c_x, 3c_x, \ldots, N_x c_x\}$, with $c_x = 2\pi/N_x$ and $Z \in \{c_z, 2c_z, 3c_z, \ldots, N_z c_z\}$, with $c_z = 2\pi/N_z$ where $N_x$ and $N_z$ are the cardinalities of each set. As both $X$ and $Z$ are discrete we can store $g, f^{\mu}, f^{\Sigma}$ as tables, hence the trainable parameters are just the function values at each point $\theta = (g_1, \ldots, g_{N_z})$ and $\eta = (\mu_1, \ldots, \mu_{N_x}, \sigma_1, \ldots, \sigma_{N_x})$. This emulates a high capacity network that can model any functional relationship between latents and observations. The prior $p(Z)$ is chosen as uniform and the coupling term is

$$p(Z'|Z=z) \quad = \quad \mathcal{VM}(Z';z,\nu_{\rho})$$

where the spread term $\nu_{\rho}$ is chosen on the order of $10^{-3}$.

In Figure 8, we illustrate an example where we fit a VAE to the empirical data distribution $\hat{\pi}(X)$. The encoder-empirical data joint distribution $\hat{\pi}(X)q(Z|X;\eta)$ and the decoder joint distribution $p(X|Z)p(Z)$ are in fact closely matching but only on the support of $\hat{\pi}$. In the next panel, we show the encoder-model data joint distribution $p(X;\theta)q(Z|X;\eta)$. The nonsmooth nature of the encoder is evident, conditional distributions at each row are quite different from one other. This reveals that samples that can be still generated with high probability with the decoder would be mapped to unrelated states when encoded again.

## E  Wasserstein Distance

The $\ell_2$-Wasserstein distance $\mathcal{W}_2$ between two Gaussians $P_a$ and $P_b$ with means $\mu_a, \mu_b$ and covariance matrices $\Sigma_a, \Sigma_b$ is given by

$$\begin{aligned}\mathcal{W}_2^2(P_a,P_b) \quad &\equiv \quad \|\mu_a - \mu_b\|_2^2 \\ &+ \mathbf{Tr}\left(\Sigma_a + \Sigma_b - 2\left(\Sigma_b^{1/2}\Sigma_a\Sigma_b^{1/2}\right)^{1/2}\right)\end{aligned}$$

## F  Experimental Details

All experiments are performed on NVIDIA Tesla P100 GPU. Optimization is performed using AdamOptimizer with learning rate 1e-4.

Figure 9: Comparison of Algorithms (on ColorMnist with Conv arch) for various different $\rho$ and $\rho_{SE}$ values. (LEFT) $\rho = 0.95$ and $\rho_{SE} = 0.975$. (MIDDLE) $\rho = 0.97$ and $\rho_{SE} = 0.97$. (RIGHT) $\rho = 0.97$ and $\rho_{SE} = .975$

**Color Mnist** For experiments with the Color MNIST dataset with MLP architecture, a 4 layer multi layer perceptron (MLP), with 200 neurons at each layer, is used for both encoder and decoder. For experiments with the Color MNIST dataset and conv architecture, a 7 layer VGG network is used for encoder with number of output channels 8, 16, 32, 64, 128, 256 and 512 respectively with strides 2, 1, 2, 1, 2, 1 and 2 respectively and kernel shape of (3, 3). For decoder a de-convolutional architecture 3 de-conv layers with output channels 64, 32 and 3, strides 1, 2 and 1, and kernel shape (3, 3) is used. Convolutional architectures are stabilized using BatchNorm between each convolutional layer. In training where adversarial attack is required, PGD with L-inf perturbation radius and iteration budget 20 is used. In PGD no random restarts are used for training, while evaluating 10 random restarts are used. In evaluation, PGD with iteration budget 40 is used.

**CelebA** CelebA experiments are performed using VGG encoder with 4 convolutional layers, output channels 128, 256, 512, and 1024 respectively, stride 2 and kernel size (5, 5). CelebA decoder is VGG network with 4 layers, output channels 512, 256, 128 and 3, stride 2 and kernel shape (5, 5). Al convolutional layers are normalized with BatchNorm. In training where adversarial attack is required, PGD attack with iteration budget 5 is used with L-inf perturbation radius. In training no random restarts are used in PGD where as in evaluation 10 random restarts are used with 20 iteration budget.

## G  Comparing Algorithms for different $\rho$ settings

Figure 9 shows comparison of all algorithms for different $\rho$ and $\rho_{SE}$ values. Figure solidifies the claim that SE-AVAE achieves better downstream adversarial accuracy than both SE and VAEA, for smaller $\epsilon$ values for reasonable $\rho$ and $\rho_{SE}$ values.

## H  CelebA Results for All Downstream Tasks

Table 3 shows adversarial downstream accuracy for all 17 downstream tasks of CelebA, compared across models VAE, AVAE, SE with 5 PGD iteration budget, SE with 20 PGD iteration budget, SE AVAE and AVAE SS. Here $\epsilon = 0.0$ represents the nominal downstream accuracy.

| | VAE | | AVAE | | SE$^5$ | | SE$^{20}$ | | SE AVAE | | AVAE SS | |
|---|---|---|---|---|---|---|---|---|---|---|---|---|
| Task | 0.0 | 0.1 | 0.0 | 0.1 | 0.0 | 0.1 | 0.0 | 0.1 | 0.0 | 0.1 | 0.0 | 0.1 |
| Bald | 97.9 | 2.1 | 97.8 | 83.9 | 97.9 | 71.0 | 97.4 | 86.5 | 97.8 | 87.0 | 97.8 | 70.0 |
| Mustache | 94.9 | 0.7 | 94.8 | 89.8 | 95.0 | 69.5 | 95.7 | 84.4 | 96.3 | 92.3 | 95.2 | 74.0 |
| Eyeglasses | 95.4 | 0.0 | 94.9 | 71.7 | 95.9 | 20.3 | 95.7 | 33.0 | 95.3 | 67.5 | 94.3 | 56.8 |
| Necklace | 87.7 | 0.7 | 88.2 | 76.7 | 88.0 | 56.0 | 88.0 | 78.9 | 86.1 | 80.3 | 87.9 | 59.9 |
| Smiling | 87.7 | 0.4 | 78.5 | 3.0 | 86.9 | 3.1 | 85.7 | 1.1 | 77.9 | 6.3 | 81.6 | 1.0 |
| Lipstick | 84.5 | 0.0 | 80.4 | 6.5 | 84.5 | 2.0 | 80.3 | 0.6 | 80.7 | 11.2 | 80.3 | 0.9 |
| Bangs | 90.3 | 0.2 | 89.7 | 36.0 | 90.8 | 19.3 | 89.6 | 27.0 | 89.6 | 45.9 | 89.6 | 22.2 |
| Black Hair | 83.5 | 0.0 | 82.5 | 31.7 | 83.4 | 23.0 | 81.4 | 31.4 | 79.5 | 33.3 | 83.3 | 26.2 |
| Blond Hair | 91.5 | 0.2 | 91.1 | 46.4 | 92.5 | 35.9 | 90.7 | 53.5 | 92.4 | 55.8 | 90.6 | 37.6 |
| Brown Hair | 78.4 | 1.0 | 77.8 | 35.9 | 77.9 | 24.1 | 80.5 | 41.5 | 82.9 | 46.0 | 78.2 | 19.3 |
| Gender | 87.6 | 0.0 | 82.6 | 5.4 | 87.8 | 1.5 | 81.6 | 0.7 | 82.1 | 10.7 | 82.1 | 0.7 |
| Beard | 85.8 | 0.0 | 83.8 | 39.2 | 85.4 | 18.3 | 85.3 | 24.3 | 86.0 | 50.1 | 84.4 | 16.3 |
| Straight Hair | 79.6 | 2.3 | 79.2 | 72.2 | 79.6 | 64.8 | 78.7 | 77.3 | 78.8 | 74.9 | 79.2 | 64.8 |
| Wavy Hair | 75.8 | 0.1 | 75.8 | 17.8 | 76.1 | 10.0 | 72.8 | 10.2 | 72.7 | 22.1 | 75.5 | 9.3 |
| Earrings | 81.3 | 0.1 | 81.0 | 60.9 | 81.0 | 26.8 | 81.3 | 55.3 | 79.5 | 64.1 | 81.2 | 24.7 |
| Hat | 96.5 | 0.2 | 96.6 | 75.2 | 96.9 | 55.3 | 96.4 | 77.3 | 97.3 | 82.7 | 96.0 | 65.5 |
| Necktie | 92.6 | 0.2 | 92.5 | 61.9 | 92.7 | 41.0 | 92.7 | 51.7 | 93.0 | 72.5 | 92.6 | 39.8 |
| Time Factor | ×1 | | ×2.2 | | ×3.1 | | ×7.8 | | ×4.3 | | ×2.2 | |
| MSE | 7203.9 | | 7276.6 | | 7208.8 | | N/A | | 7269.2 | | 7347.3 | |
| FID | 99.86 | | 97.92 | | 98.00 | | N/A | | 109.4 | | 99.86 | |

Table 3: Adversarial test accuracy (in percentage) of the representations for all of classification tasks on CelebA.