[Reviews · NeurIPS 2020]

Review 1

Summary and Contributions: The authors observe that VAEs do not correctly encode samples generated by the decoder, and therefore iteratively encoding and decoding images leads to diverging behavior. This is related to adversarial robustness. This paper proposes a new method for training VAEs that does not explicitly target robustness, and yet yields better robustness on classification tasks in ColorMNIST and CelebA.

Strengths: This work seems to be theoretically well grounded, with rather convincing experiments to back up the claims. As far as I know (I'm not following closely the adversarial robustness line of research), this work is novel. The problem explored in this paper is interesting to the community, and relevant for NeurIPS.

Weaknesses: One of the underlying points is that VAEs don't have the autoencoding property, which is deemed important for robustness in representation learning. I believe this should be made clearer, as it is now a bit too obfuscated in Section 2. More in general, the theory part should include a higher-level overview of the main points and propositions. As it is, it might be hard to follow, especially because the notation deviates from the standard current notation in VAE literature. It would be nice to include a discussion about the chosen architecture, as results might strongly depend on this. For example, the role of batch normalization is not necessarily clear, and various regularization strategies (dropout, L2 regularization, observation noise in the input) might significantly affect results. It's also unclear why (according to appendix) the convolutional decoder should be much smaller than the encoder. The effect (if any) of these modifications to the model and training objective on the test set ELBO could be reported. Especially since (probably because the encoder is more constrained than in a VAE, according to the authors) robustness seems to come at the cost of decoder quality. Quantifying this and investigating it a bit further would make the paper stronger. Ideally, the experimental section should include intuitive visualizations of the tables. It would be a good idea to include experiments on a standard benchmark for generative modeling (e.g. binary MNIST or CIFAR10) using the same likelihood function as in the literature, and report the ELBO. This way, one could make sure that the trained models are reasonably good, and the experimental conclusions would be sounder.

Correctness: Yes.

Clarity: The paper is well written, although it could use some proof-reading for polishing.

Relation to Prior Work: I believe this is a major issue in the current version. The paper presents related work that is used for experimental comparison. However, only few more loosely related works are mentioned throughout the paper, and there is no dedicated related work section.

Reproducibility: Yes

Additional Feedback:


Review 2

Summary and Contributions: This paper discusses the propensity for (Gaussian) Variational Auto-encoders to drift during subsequent encodings; that is, given an original code z (or, similarly, a point x drawn from data), this paper describes the distribution of z', the decoding and then encoding of z. This paper demonstrates 1) that || z' - z || is generally non-trivial, 2) that results may be found in the analysis of this behavior, and 3) that in attempting to remove this behavior learned representations are more robust to specific adversarial perterbations for later prediction tasks. Similar to the original VAE, the authors define a latent variable model from Z -> X, this time considering pairs of observations (Z,Z') -> (X,X') with "linking coefficient" $\rho$ between the Z and Z' instances (Z ~ Gaussian, and \rho is exactly covariance between corresponding indices of Z and Z'). They then shrink estimated models toward their prior model, which at a high level is similar to the original VAE ( KL[Q || P]] penalty, but using a repeated encoding instead of a single Z ). In practice this is an encoder and decoder pair, with original VAE penalties plus penalties based on re-encoding of the output X' to Z'.

Strengths: In general I found this paper well written and pleasant to explore. While the mismatch between iterates of autoencoders is not an entirely novel concept in general (see, e.g. [1]), it is certainly a fresh perspective and in my opinion a novel analysis of the noted phenomenon, specifically for VAE. The derived model is intuitively similar to the original VAE once it is understood, yet expressive for the purpose of exploring auto-encoder drift. Even from the latent variable/generative perspective such as it is, in my opinion this paper includes interesting work in VAE theory. It further introduces clever tools for describing and testing drift phenomena. I think the resulting simplicity of the derived model (modulo the original VAE training) is understated. That the end result separates so cleanly into the same KL[q||p] and entropy terms plus a Z -> Z' transition likelihood under a known distribution is incredibly elegant in my opinion.

Weaknesses: I found the details of the reasoning fairly difficult for sections 2 and 3; while the individual statements are meaningful and well explained usually, the reasoning for introducing or building Prop 2.3 and 3.1 is not immediately clear, even though it eventually becomes critical to the construction of the actual proposed method. Neither the original problem nor the adversarial attacks in the empirical experiments are particularly well motivated from the perspective of the theoretical analysis. The existence and nature of VAE drift is somewhat left to the reader, and its link to adversarial vulnerability is overall unexplored. The empirical results have an interesting set of experiments on adversarial robustness, but it seems important to explicitly link these experiments more closely (even just in intuition, with waving hands) with what appears to be the main idea of the paper, the analysis in Sections 2 and 3. Similarly, while I understand it is not the main focus of the work, results relative to the smooth encoder method are somewhat underwhelming without context or analytic comparison.

Correctness: Results and derivations appear correct.

Clarity: Generally clear, notation with $\theta$ and $\eta$ should maybe require a second look (is it truly necessary?).

Relation to Prior Work: The adversarial experiments are (again) perhaps undermotivated, and it is not explicitly stated what relation either AVAE or compared methods share (except somewhat in the supplement). Other auto-encoder inconsistency studies are not referenced (e.g. [1]), though the literature is sparse. [1] Alain, Guillaume, and Yoshua Bengio. "What regularized auto-encoders learn from the data-generating distribution." The Journal of Machine Learning Research 15.1 (2014): 3563-3593.

Reproducibility: Yes

Additional Feedback: 1) Figure 1 does not aid your case that this phenomenon exists. While I am familiar with this line of research from other work and thus know that VAEs and other encoders drift, a better argument should be made to the reader, both in the introduction and in Figure 1. 2) The result that the AVAE condition makes representations more robust is empirical. This isn't a problem (empirical results are good too), but it seems almost independent to the theoretical frame and intuition of the work. I understand there are space constraints (this is a very full paper), but an analysis of why VAE vulnerabilities to adversarial attacks are mitigated by the AVAE condition would be helpful. Notably, $\varepsilon$ perturbation type attacks are obviously not constrained to any data manifold (or, in a probabilistic sense, may move data $x$ to rare events $x + \varepsilon$). It is not immediately clear why a data-amortized or $z$-amortized condition should result in robustness to non-data-manifold points $x + \varepsilon$. This is maybe similar to 1), in that it's not clear to the reader why we should think about AVAE for this problem (or alternatively, why drift in autoencoders allows for adversarial vulnerability). 3) The data distribution in Eq. 5 is an isotropic Gaussian. As noted earlier, the original VAE paper doesn't really specify $p(X|Z)$, but could be one of many (named, learned, etc.). Does this also apply to the results here? I think it does (otherwise section 3.1 is irregular), but this should be noted in Section 3, adjacent to equation 5, and verified in the supplement that you haven't used specific properties of $p(X|Z)$ in any proof. The final statement of Prop. 3.1. would also need to be changed, as v appears to be a conditional Gaussian specific parameter. 4) Is there a notion of "natural drift"? In the prior $\bar{\mathcal{P}}$ the smoothness parameter $\rho$ is left as just that, a parameter. Deriving a posterior for that parameter is a big ask, but one might ask: does an estimated $\rho$ from a regular (non-AVAE) Gaussian variational auto-encoder tell us something about our data? Is there some understanding to be gained from auto-encoder drift/mixing times? Or is it simply an undesirable property similar to overfitting or mode-collapse in GANs, etc., to be removed by regularization. Or does the degradation of MSE performance vs. vanilla VAE imply that the AVAE is relatively misspecified vs. the original model, and that Z does not have "natural smoothness"? Along the same lines, why is $\rho \approx 1$ desirable is all cases? 4.1) In my opinion, since the example 3.1 is mostly relegated to the appendix, it would be helpful instead to have further experimental results; instead of focusing on adversarial experiments, testing $\rho$ w.r.t. MSE performance seems interesting. This also seems important since these results are referenced in the conclusion without supporting evidence. 5) In Section 3 notation switches to $p_\theta$ to denote a decoder held constant. Proposition 3.1 is used as justification for this; I do not understand the connection. It does not seem to affect the actual objective, but instead necessitates the `stopgradient` in implementation. Justification is unclear here. 6) Again, I think the resulting simplicity of the derived model is understated, and the end result is incredibly elegant. Unless this is obvious to others, I think it would benefit potential readers to know at the start that such a result is present, and that the derived training desiderata have the same simplicity (both in theory and implementation) as the original VAE model. Minor notes: - Equation at end of page 2 (prior to line 69) is missing a number. - Proposition 3.1 makes a statement about "v" which is only previously used in equation 5. It would be helpful to have a backreference at 3.1 (perhaps in text preceding the prop.), since "v" only appears in that definition, and is never referred to in the main text. - Figure 1 Caption: Mnist -> MNIST - Line 149 effecting -> affect


Review 3

Summary and Contributions: [====After rebuttal==== I have read your rebuttal. Thank you for addressing the concerns. Overall, I think this is a good paper and I am happy to keep my score as it is.] ------------------------------------------------ This paper investigates the inconsistency between encoder and decoder in VAE where the encoder cannot re-encode the decoder’s generated samples. The paper then proposes a self-consistency method to fix this issue by using an alternative construction of the variational distribution. The resulting objective is intuitive in that it explicitly enforces the consistency between the representation z used to generate a sample and the encoding of the generated sample. The paper conducted experiments in discrete data to show the consistency achieved by their proposed model, and also empirically show that the proposed model is more robust to adversarial attacks in colorMNIST and CelebA.

Strengths: There are several things to like about this paper: - Well motivated and well written - A principled derivation of the proposed self-consistency objective - A promising experiment where adversarial robustness can be achieved via self-consistency and without adversarial training signal.

Weaknesses: The experimental results are obtained in relatively simple data (discrete data, ColorMNIST and CelebA). Especially, the experiment that shows the consistency of AVAE is conducted only in discrete data. It is unclear where the consistency and adversarial robustness of AVAE can translate into more “realistic” data such CIFAR10.

Correctness: The derivations in the paper appear correct.

Clarity: The paper is very well written and explained.

Relation to Prior Work: The paper well refer to the related work. For example, in line 187, the paper clearly mentions that the target model is already used in [4].

Reproducibility: Yes

Additional Feedback: Line 174: Extra “.” should be moved up to the previous equation I think Fig.1 (or at least the way Fig. 1. Is presented) is not strong enough to support the evidence about the inconsistency of the nominal VAE. Since P(Z|X = number 9) is a distribution, it happens (though with low but nonzero probability) that the latent sample z ~ P(Z|X = number 9) could be in the latent region where it generates number 4 most of the time. A framework fixes this problem means that it should well separate the latent space. But P(Z|X) is a Gaussian, given any x, for any z, P(Z=z|X=x) > 0. To improve this, the paper might emphasize on the aforementioned point or might say something along the line that the generated sequence in Fig.1 is obtained with high probability. The legend in Fig.5 can be written more clearly. E.g., should move “(Top row)” near MLP architecture and “(Bottom row)” near convnet architecture, and put “and” between them. In the first column of Fig.5, why the adversarial accuracy reduces for \rho very close to 1? Fig. 1 shows the inconsistency between encoder and decoder of VAE and motivates AVAE. I expected that there would be a similar experiment to show that the proposed AVAE has the consistency between encoder and decoder but the paper never mentioned this. The only experiment showing the consistency of the AVAE is Fig.3 for discrete data. I think it would be interesting to show the consistency of AVAE in the same manner as in Fig. 1 for MNIST.


Review 4

Summary and Contributions: The authors propose a novel objective for VAEs that improves the adversarial robustness of the learnt representations. Compared to the recent "Adversarially Robust Representations with Smooth Encoders" paper (SE) [4], their formulation does not require to find adversarial examples during training which makes training faster. The authors also show that their approach is complementary to [4] and propose combination of their models with SE.

Strengths: This paper proposes a novel VAE object which is theoretically sound and clearly motivated. They improve over [4] in terms of training performance. The analysis they perform brings novel insights on VAEs and can be of a broader interest for the community. All claims are proved and valid.

Weaknesses: I would advise to make explicit in the title that this model focuses on learning adversarially robust representations using VAEs. Also, I think that the first observation "We uncover that widely used VAE models are not autoencoding - samples generated by the decoder of a VAE are not mapped to the corresponding representations by the encoder" was already made in [4].

Correctness: Yes. The empirical methodology is inspired from [4] and clearly compares with existing methods.

Clarity: Absolutely.

Relation to Prior Work: Maybe insisting on the fact that this provides an alternate approach/extension to [4] in the introduction could make the introduction and the claims more impactful.

Reproducibility: Yes

Additional Feedback: -"In this paper, our starting point is based on the assumption that if the learned decoder can provide a good approximation to the true data distribution, the exact posterior distribution (implied by the decoder) tends to possess many of the mentioned desired properties of a good representation, such as robustness." seems in contradiction with what is done in the following of the paper. -Why is X' depending on Z' in Eq. 2 and throughout Sect. 2 while it is no longer the case in Sect. 3? -Beginning of Sect. 3, it could be interesting to insist on the meaning of X tilde (all possible images) and on the distribution u tilde whose choice is possible since the image manifold is compact. Typos in appendix: - l.412 double bar for KL - l.415 why q tilde for the marginals?

[Author Response · NeurIPS 2020]

We thank all reviewers (**R1**, **R2**, **R3**, **R4**) for their detailed and encouraging comments, and we are pleased that the presentation was clear and the the work is overall well motivated. We tried to answer concerns raised by the reviewers below.

**(\*) The relation between the autoencoding and adversarial robustness is not clear(R1, R2), More intuition is needed about how the theory in 2. links with the proposed approach in 3. (R2), The results are empirical and disconnected from theory (R2).** Learning a VAE is a highly over-complete and ill-posed problem. Our intuition is that we need regularization, but instead of modifying the decoder model, we constrain the encoder, by explicitly enforcing properties of the exact posterior. Our argument starts by proposing an extended model in 2.1 that admits VAE as a marginal. In 2.2, we say that the exact decoder model is by construction independent of the choice of the coupling strength $\rho$; 2.2 simply means that, if we had access to the exact decoder and if we 'could' do exact inference, any $\rho$ will work, so the extended model is actually redundant, and in 2.3, we highlight the properties of an exact encoder. However, as we will be learning the decoder from data while doing only approximate amortized inference, we want to retain the properties of the exact posterior of the extended model (as it is more relevant for the representation learning Example 2.2), but how do we bake this in explicitly to the encoder? The answer comes in 3, where we propose AVAE, and in 3.1 we provide the justification that this choice coincides with the exact target conditional for $\rho = 1$, the representation learning case, when we encode and decode consistently.

How all this is related to adversarial robustness? The existence of 'surprising' adversarial examples, (where a pig is classified as a plane by slightly changing pixels), is typically the result of a problem with the smoothness of the representation (e.g. having a large Lipschitz constant), (see also example 3.1, VAE case). Authors in [4] attempted to fix this by data augmentation while training the encoder. We show here a more general framework, (where [4] is also a special case) and investigate an orthogonal choice that circumvents adversarial attacks. The data augmentation is achieved by using the learned generative model itself, as a component of the encoder. The AVAE objective ensures that samples that can be generated by the decoder in the vicinity of the representations corresponding to the training inputs are consistently encoded. In the experimental section we illustrate that this translates to a nontrivial adversarial robustness performance. We don't have formal guarantees, but in our opinion, learning an encoder that retains properties of an exact posterior is key in achieving adversarial robustness.

**Discuss the effect of the architecture on the results, Report ELBO for comparison (R1)** We find batch-norm useful for speeding up training while avoiding degenerate solutions, in especially VAEs that have powerful decoders. We also find that we need to choose decoders that are shallower than the encoders. Additionally in this paper, our focus was on representation learning, hence our evaluations were based on downstream tasks. We consider it a future work to investigate the issue of learning a good quality decoder (as measured by ELBO, MSE or the FID scores) using the AVAE objective alone and we conjecture that it is feasible to learn a better decoder while learning a robust encoder. Instead of the ELBO, we report the MSE as a proxy for the decoder quality as the AVAE or SE Elbo have additional terms that makes direct comparisons difficult.

**More realistic data, such as CIFAR10 (R3)** This is certainly a valid critique, and we agree that a 'wide domain' dataset such as CIFAR10 in contrast to 'limited domain' datasets MNIST (hand written digits) or CelebA (faces) are much more challenging. On the other had, our experience is that training VAE's for CIFAR10/CIFAR100 requires more advanced architectures choices, such as ResNets, or other improvements, such as VQ-VAE.

**Title change (R4)** This is a valid suggestion that we will consider; in fact our original title was 'Robust Representations with the Autoencoding Variational Autoencoder'.

**Missing related work (R1, R2)** The page limit has not allowed us to include a separate related work section but we will include more citations in the introduction and conclusion. Including Alain and Bengio (**R2**). Most work deals with modifying the generative model $\mathcal{P}$ but the spirit of our approach is regularization of the approximating distribution $\mathcal{Q}$ on an extended space.

**Additional feedback (*R2*) 1)** *Figure 1 is not useful* The drift is a consequence of the inconsistency of the encoder and decoder, even if we choose $\rho = 1$, please see above (\*) *(R2)* **4)** *Learning a natural $\rho$ coupling parameter?* In the standard VAE data is assumed to be iid, and as Proposition 2.2 also shows, the marginal is independent of $\rho$. Hence this parameter is not identifiable from data, unless we assume additional relational structure, such as in a video where subsequent frames are closely related. But this requires observing at least pairs of data points which is actually not available in the standard benchmarks. *(R2)* **4.0)** *Why $\rho = 1$?* This is a hyper parameter that can be chosen freely (VAE implicitly has $\rho = 0$) Any choice close to one is reasonable in the context of representation learning; in lack of any other downstream task we would like to retain a representation that would enable us to reconstruct the same image. In our experiments, we tune this parameter and find $\rho = 0.95$ gives good results. *(R2)* **5)** $p_\theta$ *notation is unclear* Our wording seems to cause the misunderstanding here; the last sentence before 3.1 should have read 'The following proposition shows our justification for the choice of $\mathcal{Q}_{\text{AVAE}}$ distribution'. The justification for $p_\theta$, does not follow from 3.1.

[Meta-Review · NeurIPS 2020]

All reviewers agree that this is a good contribution to the VAE literature. My recommendation is to accept. Please take the reviewers' comments into account in preparing the final version of the paper.